# A Morphological and Ultrastructural Study of the Anterior Digestive Tract of Adult Nile Tilapia *Oreochromis niloticus*

**DOI:** 10.3390/ani13030420

**Published:** 2023-01-26

**Authors:** Antonio Palladino, Elena De Felice, Chiara Attanasio, Carmela M. A. Barone, Antonio Crasto, Livia D’Angelo, Daniela Giaquinto, Claudia Lambiase, Paola Scocco, Francesco Serrapica, Lucianna Maruccio

**Affiliations:** 1Department of Agricultural Sciences, University of Naples Federico II, Via Università 100, 80055 Portici, Italy; 2School of Biosciences and Veterinary Medicine, University of Camerino, Via Gentile III da Varano, 62032 Camerino, Italy; 3Department of Veterinary Medicine and Animal Production, University of Naples Federico II, Via F. Delpino 1, 80137 Naples, Italy

**Keywords:** *Oreochromis niloticus*, esophagus, stomach, morphometry, conventional glycohistochemistry, SEM

## Abstract

**Simple Summary:**

Aquaculture is currently one of the fastest growing food-producing systems due to its relevance as source of livelihood worldwide. As fishes live in very diversified environmental conditions and this diversification also impacts their diet, it is not surprising that a considerable variability exists in the organization of the digestive tract. Nile tilapia is among the most-used fish species in aquaculture due to its growth and ability to adapt to a wide range of growing conditions. The aim of this work is an integrated description of esophagus and stomach morphology. We identified five distinct zones of the stomach via light and scanning microscopy. Histochemical investigation showed the presence of carboxylated and sulphated mucins along the esophagus and stomach capable of attracting a great amount of water and enhancing the protective function of the mucosa mucous gel, especially in the highly acidic stomach environment that digests algae and detrital bacteria. Therefore, our study provides new insights concerning the morphological structure of the anterior digestive tract of Nile tilapia, relevant to better understand the several aspects related to physiological growth and the health status of this species, while at the same time comparing different or pathological conditions.

**Abstract:**

Among the most-used fish species in aquaculture is the Nile tilapia, due to its rapid growth rate and its adaptation to a wide range of farming conditions. A careful description of the morphology of the digestive tract, particularly the esophagus and stomach, allows a better understanding of the relationship between structure and function. Combining scanning and light microscopy we highlighted the presence of five different zones in the stomach (1: esophagus-gastric lumen passage; 2: descending glandular portion; 3: fundic portion; 4: ascending glandular portion; 5: gastric-pyloric transition portion). Histochemical investigation showed a secretion of carboxylates mucopolysaccharides along the esophagus and sulphated complex carbohydrates in the stomach. These results suggest that mucins play a protective role of the epithelial lining, which is essential for a correct digestive process. Finally, the characterization of the main cellular structures may be inspiring for more advanced studies aiming to decipher the role of specific molecules, such as neuropeptides, involved in the physiological digestive process.

## 1. Introduction

Aquaculture is currently one of the fastest growing food-producing systems due to its relevance as a source of livelihood worldwide. In commercial farming, fish exposed to suboptimal environmental conditions can alter feeding behavior and growth rate as well as energy metabolism [1]. In addition, fish living in highly diversified aquatic environments due to the different types of food available will display morpho-functional variations of the digestive tract, mainly related to the stomach [2,3]. Indeed, this organ may be even absent; when present, it may show different shapes [4,5] or be modified as a swelling structure (called intestinal bulb) [6,7,8,9,10,11]. The extreme variability of stomach shape and structure in Teleosts is mainly related to phylogeny and ontogenesis, but also to functional adaptation [12,13,14,15,16]. Compliant with different feeding habits, carnivorous fishes feature large, extensible stomachs with a short intestine [9,10,11,17,18], while omnivorous fishes display a large stomach and a long intestine [19,20]. Interestingly, strictly herbivorous fish feature a small stomach with a long and complex intestine, but in some cases the stomach may be completely absent [14,18,19,21,22,23,24,25,26,27]. As the morphology of the digestive system of fishes is strictly related to specific diet, correct management of feeding habits is mandatory to improve production rates of fish.

Of the fish species used in aquaculture, Tilapia is the third most-used after carp and salmon [28,29]. It is an omnivorous fish of the Cichlid family which lives in Africa, South America and Asia. Nile tilapia (*Oreochromis niloticus*) is one of the most farmed tilapia species, due to its rapid growth rate and ability to adapt to a wide range of environmental conditions [28]. Tilapia is appreciated by consumers for the softness of its muscles, its low fat content and good flavour [29]. This is a tropical species living in shallow and sweet waters (lakes and lacustrine areas) with temperatures ranging from 31 to 36 °C; however, Tilapia is resistant to variations of these environmental parameters. It can live up to 10 years and reach a weight of about 5 kg [28]. 

Very few studies describing the ultrastructure of the digestive tract of Nile tilapia are reported in literature [30,31,32] and only one shows a figure of the entire stomach stained by Masson’s trichrome [33]. Although the basic histological structure of the digestive tract is similar in the different fish species [34], significant differences can be observed macroscopically in the different regions of the stomach. In particular, the differences highlighted in wall architecture and mucosa ultrastructure of the digestive tract of Nile tilapia [30,31,32] determine different secreting behavior and mucosa architecture of each one of the different regions of the stomach. 

The shape of the stomach and the topographical proximity between the cardia and pylorus probably influences food transit; therefore, also influencing the morphological and functional features, such as enzyme secretion.

The aim of this research is an integrated description of esophagus and stomach morphology of Nile tilapia by combining scanning electron and optical microscopy, highlighting the different possible distributions of mucopolysaccharides to better understand the relationship between structure and function.

## 2. Materials and Methods

### 2.1. Sample Collection

For this study, ten adult female Nile tilapia *O. niloticus*, with an average weight of 200 g and 20 cm long, were used. These fish, intended for human consumption, were obtained from a local farmer in the north of Italy. The animal care procedures were in accordance with legislative decree No. 146, which implements directive 98/58/EC, 20 July 1998, concerning the protection of farmed animals. After a 24-hour fast, the fish were suppressed at lower temperature. Then, samples of the esophagus and stomach were removed through a ventral incision of the body from the anal opening to the operculum, washed in phosphate buffer (PBS) pH 7.4 and fixed as follows: 

1. in Bouin’s liquid for 24 hours (h) at room temperature (RT) for histological and histochemical analyses; 

2. in 2.5% glutaraldehyde solution in PBS pH 7.4 for 24 h at 4 °C for scanning electron microscopy (SEM).

### 2.2. Morphological and Conventional Glycohistochemical Analysis 

Whole stomachs with the respective esophagus were fixed and then dehydrated in graded ethanol concentrations (70% to absolute), clarified through two baths in xylene and embedded in paraffin wax at 56–58 °C under a thermostat-vacuum paraffin-embedding bath for 1.5 h. Sagittal sections of 7 μm thick were obtained by using a Reichert–Jung 2050 microtome, then mounted on clean and dry glass slides. Sections were stained with Harris’s hematoxylin-eosin (HE) for morphological evaluation. For histochemical analysis, samples were treated with periodic acid Schiff (PAS), thereby visualizing the vicinal hydroxyls groups; Alcian blue (AB) pH 2.5, thereby visualizing acidic groups pertaining to carboxylated and sulphated complex carbohydrates (sialylated glycoproteins, hyaluronic acid, chondroitin, chondroitin-sulphates A/B/C, heparin, heparansulphate glycosaminoglycan-like material); AB pH 1, thereby visualizing acidic groups pertaining to sulphated complex carbohydrates (chondroitin-sulphates A/B/C, heparin, heparansulphate glycosaminoglycan-like material); and periodic acid Schiff in combination with the Alcian blue (AB/PAS), thereby co-visualizing acidic groups and vicinal hydroxyls [35,36,37]. Staining solutions were purchased from Bio-Optica Milano SPA: HE (code 05-M06004); AB pH 2.5 (code 05-M26003); AB pH 1 (code 05-M26005); PAS (code 04-130802A); AB/PAS (code 04-163802). All reactions were carried out according to the manufacturer’s instructions. 

All stained sections were observed and photographed (ten for 20× for the mosaic and 40× for the particulars) under Leica DM 1000 light microscope. The digital raw images were optimized for image resolution, contrast, evenness of illumination and background using Adobe Photoshop CS5 (Adobe Systems, San Jose, CA, USA).

### 2.3. Morphometry

The thickness of the superficial and glandular epithelium, submucosa, and circular and longitudinal musculature of the stomach were measured by digital imaging and Image J software. For each sample, a total of 3 sections have been processed and 10 random microscopic fields were analyzed at 20 × magnification. In particular, the thickness was calculated as the average of four measurements. In order to highlight statistically significant differences among the same layers in different zones, we performed the statistical analysis described below. The analysis consisted in a one-way ANOVA test on each mucosal layer. This test was conducted in multiple comparison within each group, posing the values of Zone 1 as “internal control”. We evaluated measurements of the same zone from 13 different slides. Through this approach, 4 *p*-values were calculated corresponding to Zone 1 vs. Zone 2; Zone 1 vs. Zone 3; Zone 1 vs. Zone 4; and Zone 1 vs. Zone 5. The differences were considered significant when the *p*-value was <0.05.

### 2.4. Scanning Electron Microscopy (SEM)

Specimens of 5 × 5 mm were postfixed with 1% osmium tetroxide in PBS for 6 h at 4 °C in the dark. After three washes in PBS, the specimens were dehydrated through a graded series of ethanol and subsequently in CO_2_ atmosphere, using a critical point dried EMITECH K850 (Quorum, East Sussex, UK). They were attached to aluminum stubs facing upwards, covered with carbon tabs, and then the samples were sputtered with gold palladium (AGARIB 7340, Agar Scientific Ltd, Stansed, UK). The sections were examined with a scanning electron microscope ZEISS EVO40 (20.000KU EHT, 10.0 mm ˂ ωD ˂ 13.0 and 90× ˂ magnification ˂ 8k).

## 3. Results

### 3.1. Gross Morphology

The esophagus appeared as a short and dilatable membranous tube connecting the pharyngeal cavity to the initial part of the stomach, displaying the same diameter along its entire length (Figure 1A). 

The stomach was similar to a U-shaped pouch, with a descending portion and an ascending portion positioned in the abdominal cavity ventrally and dorsally, respectively (Figure 1A).

### 3.2. Microscopical Organization

The esophagus and the stomach showed the typical stratigraphy of hollow organs consisting of four layers or tunics; from the external layer to the lumen these are: the serous; the double muscle layer (inner circular and outer longitudinal), composed of striated fibers; the submucosal and the mucosal. In particular, the muscle layer displayed a different thickness along the stomach, allowing us to identify up to three layers in the regions featured by the thickest wall. These layers were arranged as follows: inner circular, middle oblique and outer longitudinal (Figure 1B).

The serous tunic of the esophagus was composed of two layers: the subserous lamina, made of loose connective tissue and highly vascularized; and the serous epithelial lamina. The inner longitudinal muscle layer was quite thick, while the outer circular layer was characterized by bundle of muscle fibers separated by loose connective tissue. The lamina propria of the mucosa was organized in a thick stratum of collagen fibers. The mucosa showed numerous longitudinal folds along the entire length of the organ, giving the lumen of the esophagus a star-shaped appearance, mainly when observed in a transversal section. These mucosa folds were covered with a layer of squamous epithelium, including mucous secreting cells (Figure 2A) featuring an elongated or a rounded shape. The rounded cells that appeared larger compared with the elongated cells were concentrated basally, while the small elongated cells were localized on the epithelium surface. Interestingly, we didn’t detect taste buds within the esophagus mucosa and the component known as *muscularis mucosae*. The transition from the esophagus to the stomach, corresponding to the cardial zone, was characterized by a cylindrical glandular epithelium (Figure 3C). 

The stomach serosa was formed by a vascularized and innervated tissue composed of a simple epithelium and a thin layer of connective tissue exerting mainly a mechanical support function. In the muscular layer, we observed a smooth musculature organized in fibers displaying a circular and a longitudinal course in the inner and outer layer, respectively. In addition, in the fundic zone, towards the lumen, we identified a third internal layer formed by diagonal fibers (Figure 3E). The submucosa architecture displayed a stratum of loose connective tissue joining the mucosa to the muscle tunica and including large blood and lymphatic vessels. We also highlighted the presence of circular structures in the submucosal layer that could be a nervous net comparable with those described in the mucosal wall of rainbow trout stomach [18]. The connective tissue allows the epithelium to slide over the underlying layers of the wall. The mucosa was characterized by two portions: one external, including the gastric tubular glands, and one internal (luminal), featuring elongated cylindrical cells with the nucleus located in the basal portion (Figure 3A–C). This layer of the mucosa was organized in folds showing basal "gastric areoles" featured by apical holes known as "gastric pits". The distal portion of the stomach, namely the pyloric zone, was characterized by a muscular layer of striated muscle, as well as a reduction, up to the disappearance, of the glandular component (Figure 2B).

Based on morphological analyses, we identified five distinct zones with different morpho-functional features in the stomach (Figure 4):ZONE 1: esophagus–gastric lumen passageZONE 2: descending glandular portionZONE 3: fundic portionZONE 4: ascending glandular portionZONE 5: gastric–pyloric transition portion

**Zone 1** corresponded to the transition area from the esophagus to the stomach. The first segment of the esophagus presented a squamous epithelium rich in mucous cells, while towards the stomach we observed an initial reduction in the number of mucous cells, then a progressive replacement of the squamous epithelium with that typical of the gastric region. A gradual increase of gastric glands was evident immediately beneath this superficial epithelium. The submucosa was made up of connective tissue, while the muscle fibers in this area were exclusively smooth. Further, in correspondence with the appearance of the population of the tubular glands, the organization of the muscular tunic included two layers, one (internal) with circular fibers and one (external) with longitudinal fibers.

**Zone 2**, descending glandular portion, showed the overlapping of the tunics of the gastric glandular portion. Here it was evident the presence of folds, including columnar cells, that covered an inner layer of glands reaching the organ lumen through the gastric pits. The submucosal layer was made up of dense connective tissue; thicker and less compact in the section corresponding to the fold, and thinner and denser in the portion at the base of the folds. In addition, circular corpuscles were observed in this layer. The smooth muscle tunic was composed of fibers arranged to form the two patterns previously described.

**Zone 3**, fundic area, was located between the descending (ventral) and the ascending (dorsal) tracts of the organ and showed an architecture intermediate to those of the two tracts. This area retained a well-developed mucosa lying superficially to the tubular glands on both sides. In contrast, the submucosal layer was different in the two tracts, appearing thin and compact in the descending tract, and thick and loose in the ascending one. At the apex, moving towards the ascending tract, there was a thickening of the muscular circular layer, while the longitudinal layer remained constant. In the muscle stratum, some transversal fibers still appeared.

**Zone 4**, ascending tract, featured both the epithelial layer and the glands. An increased height of the connective layer of the submucosa was observed in this area. At the same time, the muscular tunic continued the progressive thickening that was observed starting from the apex of the fundus. In addition, in this area the thickening was due to the internal fibers with a circular pattern, as reported for Zone 3.

**Zone 5**, gastric pyloric, the transition area between the proper gastric and the pyloric portions. The various layers appeared highly modified in this area. Starting from the epithelium, we witnessed a gradual change towards the lining of the intestinal mucosa. The tubular glands gradually disappeared, and the submucosa changed towards the pyloric sphincter. The muscular layer increased considerably in thickness, becoming the predominant layer in the proximity of the pyloric sphincter. In the area immediately preceding the pylorus, striated muscles appeared. 

### 3.3. Morphometrical Analysis

We measured the thicknesses of the different layers composing tilapia stomach by evaluating superficial and glandular epithelia, epithelium, submucosal, circular and longitudinal muscle components. The results are summarized in the diagrams shown in Figure 5. 

The mean thickness of the lining epithelium in zone 1 was 42 µm; it was significantly higher in the other zones: Zone 2, 175 µm; Zone 3, 137 µm; Zone 4, 81 µm; and Zone 5, 127 µm (*p*-values: <0.0001, <0.0001, <0.0001 and <0.0001, respectively) (Figure 5A). The glandular epithelium was uniform in Zones 1, 3 and 4 (188 µm, 165 µm, 172 µm, respectively) (*p*-values: 0.2320 and 0.5095). We measured a significant increase in Zone 2 (246 µm ) (*p*-value: 0.0001), and a significant decrease in the gastric–pyloric transition region (60 µm ) (*p*-value: 0.0001) (Figure 5B). The submucosa showed a constant thickness in the first three zones: 1, 2 and 3; 58 µm, 117 µm, and 87 µm, respectively, and a significant increase in Zones 4 and 5 (1676 μm and 417 μm, respectively) (*p*-values: <0.0001, <0.0001, respectively) (Figure 5C). The circular muscular layer in Zone 1 was 43 µm thick, showing a gradual increase in Zones 3, 4 and 5 (65 µm, 128 µm and 235 µm, respectively); however, this increase is significant only in zones 4 and 5 (*p*-values: <0.0001, <0.0001). Conversely, we detected a slight reduction in Zone 2 (20 µm) that significantly differs from Zone 1 (*p*-value: 0.0374) (Figure 5D). By contrast, the thickness of the longitudinal layer of the muscular tunic was generally uniform (32 µm, 15 µm, 25 µm, 26µm and 45µm) (*p*-values: 0.2900, 0.6870 and 0.3614), except in Zone 2, where it significantly decreased (*p*-value: 0.0001) (Figure 5E).

### 3.4. Histochemical Analysis

The histochemical investigation showed differentiated results among the treatments carried out. Moderate positivity was evidenced in all the histochemical reactions at mucous cell level in the esophagus epithelial lining, with the exception of AB pH 1.0, which displayed a negative reaction. In the stomach, the epithelial cell coat showed an intense staining to PAS, and AB/PAS (Figure 6A,D and Figure 6G, respectively), while moderately positive to AB pH 2.5 and negative to AB 1.0 (Figure 6B,E and Figure 6C,F, respectively). The gastric pits proved to be the most reactive stomach structures; in fact, they were strongly positive to PAS, AB pH 2.5 and 1.0 (Figure 6D,E,F), and prevalently AB positive to AB/PAS treatment, showing a blue staining (Figure 6G). Gastric glands showed a strong reactivity to both PAS (Figure 6A,D) and AB/PAS (Figure 6G), to which they were prevalently PAS positive, showing a pink staining. Gastric glands showed a slight AB pH 2.5 reactivity (Figure 6B,E), while they were negative to AB pH 1.0 (Figure 6C,F).

### 3.5. Scanning Electron Microscopy Obsevations

Through scanning electron microscopy (SEM) we observed the esophagus and stomach in three dimensions. 

In cross-section, the esophagus was tubular (Figure 7A), and the four layers were distinguishable from outside to inside as follows: serous, muscular with longitudinal (outer) and circular (inner) fibers, a very thick submucosa, and a mucosa featuring folds of different sizes.

We further identified five different areas of the stomach, consistent with what we observed via light microscopy. The transition area between the esophagus and the first part (Zone 1) of the stomach was characterized by the appearance of a columnar epithelium laying on a basal lamina, below which the submucosa was also evident. In addition, we noticed gastric pits with gland outlets (Figure 7B). Also, through this technique we observed that the mucosa of the descending glandular portion (Zone 2) featured numerous folds. The columnar epithelium laying on the submucosa was very evident. In this region, the glands with their outlets appeared surrounded by epithelial cells (Figure 7C). The fundic portion (Zone 3) did not differ from Zone 2. Therefore, this portion was characterized by an increase of the glands highlighted on the epithelial surface by numerous gastric pits (Figure 7D). The ascending glandular portion (Zone 4) began to show significant changes, mainly due the increase of connective tissue and, therefore, to the presence of very large mucous folds (Figure 7E). Finally, the last portion, transitioning between the properly glandular gastric portion and the non-glandular pyloric portion (Zone 5), was characterized by morphological modifications typical of the intestinal epithelium. A mucosa that is no longer glandular was clearly visible, as well as an increased muscular component, typical of the pyloric sphincter.

## 4. Discussion

In this study we described the esophagus and stomach of Nile tilapia to highlight their different cellular distribution, their morphological structure through scanning electron microscopy, and to characterize their complex carbohydrates through conventional glycohistochemical stainings. Our observations allow us to relate each morphological difference to a specific function.

Regarding the esophagus, our SEM data highlighted a mucosa architecture characterized by thin longitudinal folds of different heights, extending along the whole organ length. Microscopically, these folds were deep and included the mucosa, thus increasing the distension capacity of the esophagus during food transport [38,39]. Further, our results corroborate the hypothesis reported in other studies [30,33], according to which the typical protective function of the esophagus from mechanical insults is played by the squamous epithelium. Histochemical treatments evidenced the production by the esophagus mucous cells of carboxylated complex carbohydrates, represented by hyaluronic acid, chondroitin, chondroitin-sulphates A/B/C and glycosaminoglycan-like material that form a viscoelastic barrier that, covering the outer epithelial layer, protects the esophageal mucosa. Intriguingly, the presence of a great number of muciparous cells (both elongated and rounded) suggests a pre-gastric digestion, postulating an additional function of tilapia esophagus in digestion, as is already reported in other fish [2].

Literature data show a subdivision in the stomach of this fish. Al-Hussaini and Kholy [40] described two zones in tilapia stomach, namely one large caecal and one pyloric; in contrast, Caceci and collaborators [32] highlighted three regions, an initial or cardiac, a middle or fundic, and a terminal or pyloric. Further studies [41,42] report that the stomach of poly-hybrid Tilapia displays an identical structure in all the regions.

However, in our work we identified for the first time, via scanning microscopy and histochemical techniques, five morphologically distinct zones in the stomach, based on the composition of the gastric wall and the architecture of the superficial epithelium. The first zone (Zone 1) showed a replacement of the stratified squamous epithelium, including mucous cells of the esophagus, with a simple columnar epithelium typically displayed in the stomach. This organization is due to the transition from a non-glandular to a glandular structure, very common in Teleosts [43,44,45,46,47]. The columnar epithelium lined the folds which basis is featured by the outlet of the gastric glands, which secrete acidic and neutral mucins, as reported by several authors [30,31,33]. Different studies showed that the active secretion of mucins and the structure of the epithelium play fundamental roles in the maintenance of the physiological microenvironment, which is essential for digestive function [18]. To further confirm this, in Zone 5, corresponding to the gastric–pyloric transition portion, epithelium morphology changes, gradually displaying similarities with that of the first tract of the intestine. 

In this study we also described a progressive increase in the glandular component of the gastric mucosa, which confirms that already found in the first tract (Zone 1); subsequently, in the descending glandular portion (Zone 2), the digestion starts continuing up to the fundic (Zone 3) as well as in the ascending glandular portion (Zone 4). The disappearance of the tubular glands in the gastric–pyloric passage portion (Zone 5) is indicative of a cessation of digestive function typical of this zone. In fact, it is reported that the glandular region of the stomach exerts digestive functions while the non-glandular region is mainly devoted to the transit of food to the intestine due to its morphology (muscular layer) and function (epithelial secretions) [40]. In addition, the folds of the mucosa delimiting all the gastric lumen in the first four areas could also play a role in delaying the passage of food through the stomach. This function maximizes the mixing of food with digestive enzymes, enhancing the digestive process and nutrient absorption, as confirmed in other studies [20,31].

Folds are closely related to an increase in the submucosal connective layer. In thicker areas, the connective tissue is much less dense, thus ensuring proper mechanical resistance to the gastric wall. The three muscular layers contribute to mixing the food and to ensure its progression towards the pylori. The smaller quantity of muciparous cells in the pyloric region, compared with other regions of the stomach, suggests a role in food retention rather than a proper digestive function. Indeed, the preservation of food in the pyloric region is probably relevant for better digestion, as already observed in other fish [48]. 

Furthermore, the presence of a crest-shaped sphincter (valve) characterized by the striated muscles between the stomach and the first tract of the intestine could avoid the transit to the intestine of food that has not undergone chemical digestion, as reported by Morrison and Wright [33].

From a morphological standpoint, the stomach of Nile tilapia displays the same cellular organization of most teleost fishes, including other cichlids [10,11,49]. The glandular cells promote digestion [42] by secreting hydrochloric acid through the oxyntic cells. Therefore, their secretion plays a key role in regulating gastric pH since, unlike what happens in mammals, fish gastric glands produce only hydrochloric acid. As reported in the literature, it seems that to promote acid lysis, as well as digestion of algae and detrital bacteria, tilapia needs an acidic gastric environment during the early stage of digestion due to the lack of adaptation to trituration [31,42,50,51]. From the histochemical point of view, the tilapia stomach shows the production of a great variety of glycoconjugates. The gastric pits secrete high-acid complex carbohydrates, represented by chondroitin-sulphates A/B/C, heparin, and heparan-sulphate glycosaminoglycan-like material, with numerous negative charges protecting the gastric mucosa from the acidic environment of the stomach. The protective action is probably carried out in two ways: by repelling the negative charges and by attracting large quantities of water to form a thick layer of gel. The cell coat is also involved in the protective function; in fact, the presence of sialylated glycoproteins in this site is evidenced by histochemical treatments. Gastric glands show the presence of neutral, and in minor measure, sialylated glycoproteins [18,32,41,52]. This finding causes us to hypothesize a role for each class of mucins that, on one hand, preserve the cellular epithelial layer, and on the other, help to ensure the right conditions for correct digestive processes. 

In addition, the morphometric analysis of the muscle layers confirms a rather peculiar organization of Nile tilapia stomach. The external longitudinal layer, displaying a constant thickness throughout the lumen of the organ, exerts a key role in food stirring. Conversely, the circular layer, which increases significantly from the apex of the stomach, ensures the progression of almost completely digested food towards the pyloric sphincter. While in the descending portions of the stomach, the food advances due to gravity as well, in the ascending tracts it must proceed against gravity, therefore requiring a force able to push it up. The increased thickness of the circular muscle layer could then be attributed to this specific function.

## 5. Conclusions

In view of the key role played by fish both in translational research [53,54] and aquaculture, our work may be of interest in different research settings. This preliminary study provides in-depth morphological data of the stomach of Nile tilapia, inspiring us to formulate a functional hypothesis. The identification of the main cellular structures within the epithelium may contribute to more advanced studies aiming to unravel the basic mechanisms of digestion in this species. In addition, our dataset may be useful for studies focused on understanding the morphological changes induced by different dietary plans and aimed at improving production rates while also increasing animal welfare.

## Figures and Tables

**Figure 1 animals-13-00420-f001:**
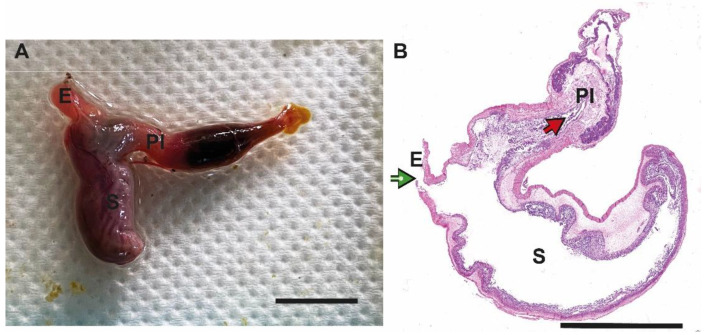
Gross morphology and microscopical organization of esophagus and stomach of Nile tilapia. (**A**): esophagus (E), stomach (S) and proximal portion of the intestine (PI); (**B**): Hematoxylin/Eosin staining of a para-sagittal section of the two organs. In the upper left is visible the esophagus outlet (green arrow), while in the upper right is the pyloric sphincter (red arrow). The figure is representative of a reconstructed mosaic of 180 individual photos taken using a 20× objective to visualize in detail the stratigraphy of the organs. Scale bars = 1 cm.

**Figure 2 animals-13-00420-f002:**
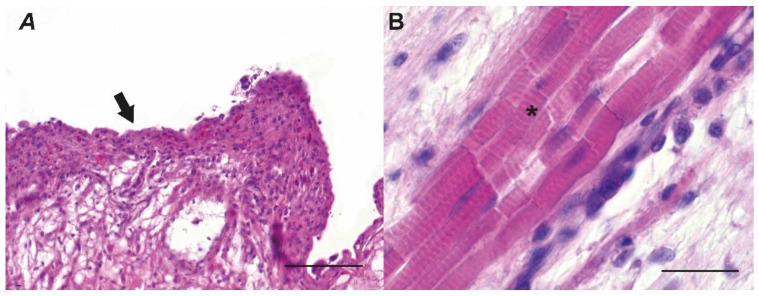
Histological characteristics of sagittal sections of esophagus walls of Nile tilapia. (**A**): mucosa folds of the esophagus covered with layer of squamous epithelium (arrow), (**B**): pyloric zone with striated muscle layers (asterisk). Scale bar = (**A**): 100 μm; (**B**): 25 μm.

**Figure 3 animals-13-00420-f003:**
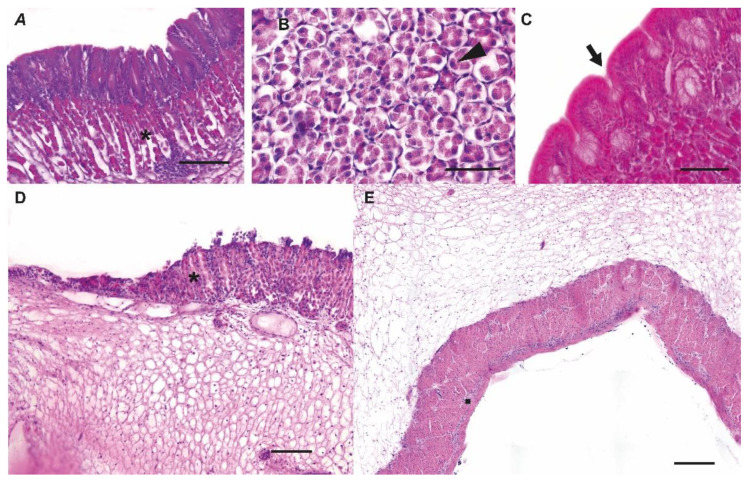
Histological characteristics of sagittal sections of stomach wall of Nile tilapia. (**A**): epithelial layer that coats the tubular glands (asterisk); (**B**) cross section of the tubular glands with prismatic cells (arrow head); (**C**): monolayer of cylindrical cells displaying the typical folds and delimiting the gastric pits (arrow); (**D**): transition from the esophageal to the gastric mucosa, showing tubular glands at the beginning (asterisk); (**E**): fundic zone (square putative third transversal muscle layer). Scale bars = (**A**,**C**,**D**,**E**): 100 μm; (**B**): 50 μm.

**Figure 4 animals-13-00420-f004:**
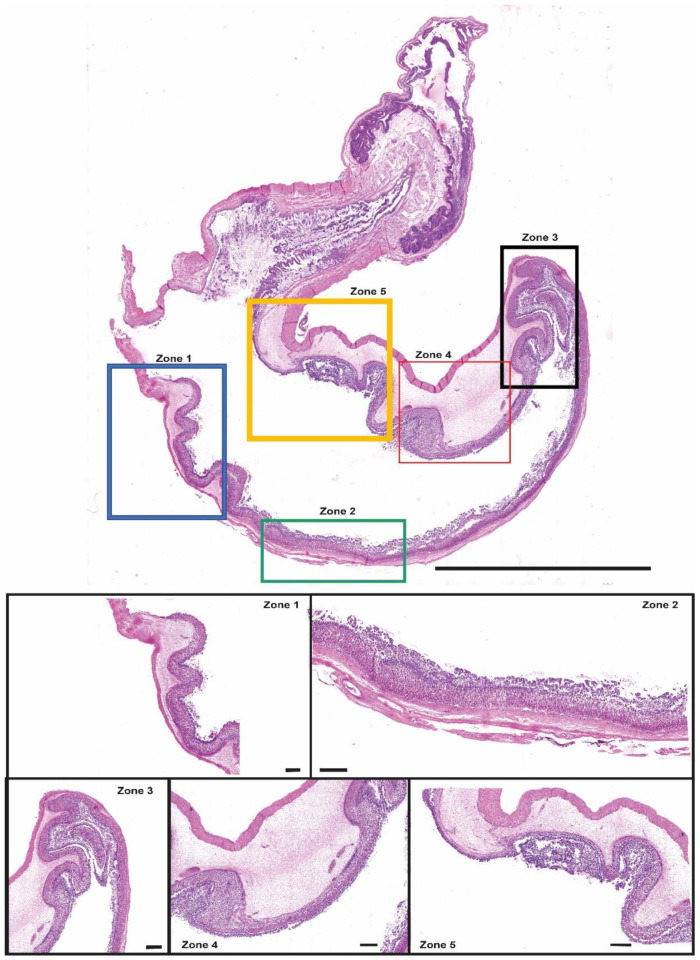
Morpho-functional organization of the stomach. Five zones based on morphological differences we highlighted: Zone 1: esophagus–gastric lumen passage, Zone 2: descending glandular portion, Zone 3: fundic portion, Zone 4: ascending glandular portion, Zone 5: gastric-pyloric transition portion. Scale bars = complete reconstruction: 1 cm; Zone 1, Zone 2, Zone 3, Zone 4, Zone 5: 200 μm.

**Figure 5 animals-13-00420-f005:**
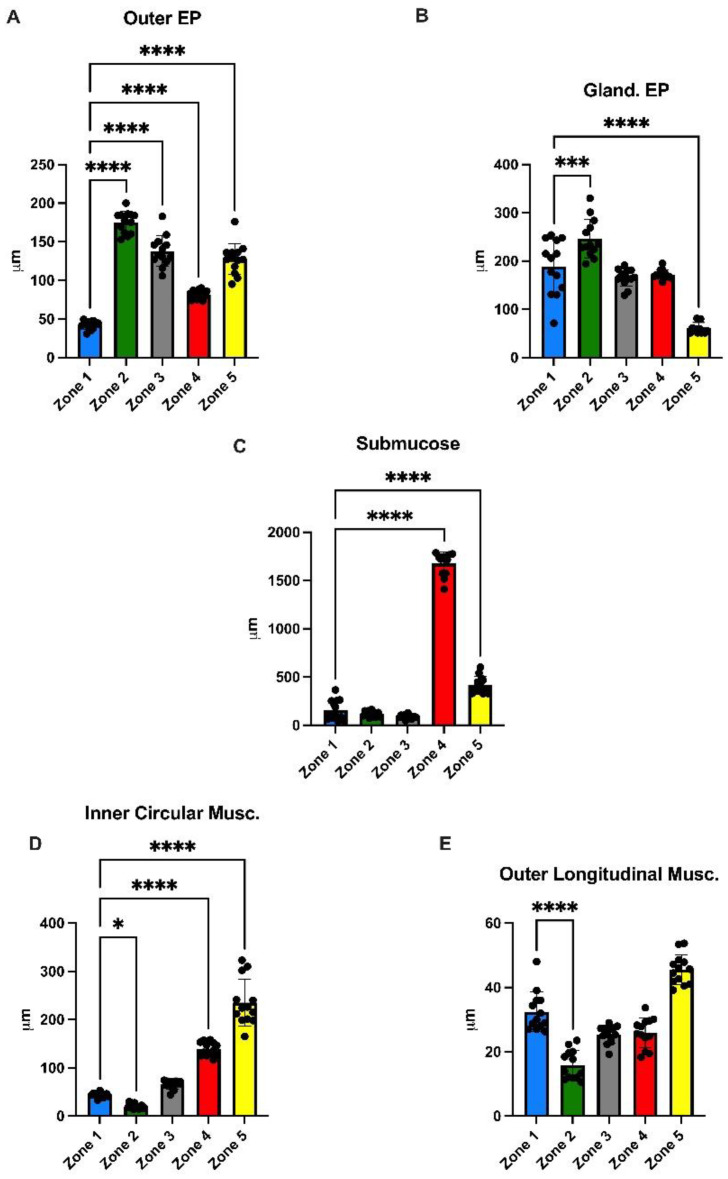
Diagrams representing the results of the morphometric analysis. (**A**): the thickness of the coating epithelium was reduced in Zone 1 and significantly more developed in Zones 2, 3 and 5, with a slight decrease in Zone 4. (**B**): the glandular epithelium, uniform in Zones 3 and 4, showed a significant increase in Zone 2 and decrease in the region of gastric–pyloric transition. (**C**): the submucosa showed a constant thickness in all zones except in Zone 4, where it increased. (**D**): the inner circular muscle layer showed a gradual increase from Zone 1 to Zone 5, except in Zone 2. (**E**): the outer longitudinal muscle layer displayed a uniform thickness, except in Zone 2 where it decreased. * *p*-values ˂ 0.05; *** *p*-values ˂ 0.001; **** *p*-values ˂ 0.0001.

**Figure 6 animals-13-00420-f006:**
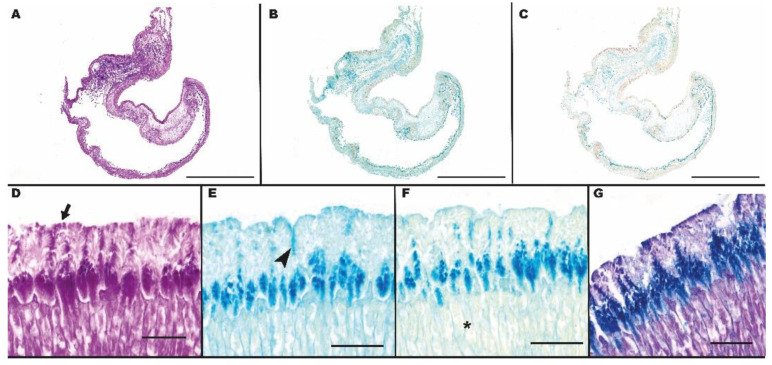
Histochemical investigation of the stomach of Nile tilapia. Figure shows periodic acid Schiff (PAS) (**A,D**); Alcian blue at pH 2.5 (**B,E**); Alcian blue at pH 1.0 (**C,F**); and Alcian blue/PAS (**G**) binding sites on mosaics of the whole stomach. Epithelial cell coat (arrow); Gastric pits (asterisk); Gastric glands (arrow head). D, E and F represent magnifications (40×) of PAS, Alcian blue at pH 2.5, and Alcian blue at pH 1.0, respectively. Scale bars = (**A**–**C**): 1 cm; (**D**–**G**): 100 μm.

**Figure 7 animals-13-00420-f007:**
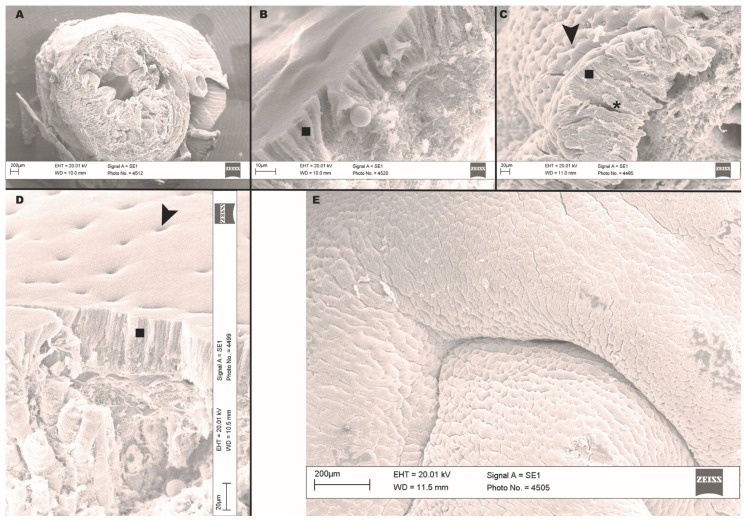
Scanning electron microscopy (SEM) images of esophagus and stomach of Nile tilapia. (**A**): cross section of esophagus. The four layers are clearly evident: serous, muscular (inner circular and outer longitudinal), submucosa and mucosa, featuring folds of different heights; (**B**): esophagus–gastric lumen transition. This section shows a single layer of columnar epithelial cells (square) and gastric glands. Vessels are evident in the submucosa; (**C**): descending glandular portion of the stomach. The outlets of the gastric pits (arrow heads) are noted, as well as the epithelial columns lining the mucosa (square) and a gastric gland (asterisk); (**D**): fundic portion of the stomach. Evident columnar epithelium (square) and gastric pits (arrow heads) laying on periglandular connective tissue (arrow); (**E**): ascending glandular portion of the stomach. Scale bars are indicated in each picture.

## Data Availability

Datasets used in the analyses are stored at the authors’ home institution and will be provided upon request from the corresponding author.

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
