# Peer review of "A Morphological and Ultrastructural Study of the Anterior Digestive Tract of Adult Nile Tilapia Oreochromis niloticus"

_animals, 2023, doi:10.3390/ani13030420_

Round 1

Reviewer 1 Report

The authors of the study showed the presence of 5 different regions in the stomach of Nile tilapia that allowed them to formulate functional hypotheses for the digestive process that could be used for studies of different dietary plans.

In general, the manuscript is interesting and I recommend it for publication after some revisions, such as: 1) the statistical analysis of the morphometric data, in order to give strength to the results; 2) the description of the histological images should be contextualized, where possible, on the images using arrow, asterisk, arrowhead, etc. in order to make reading the results clearer.

Abstract

Line 29: “growth”: higher growth rates?

Line 30: avoid repeating the word “growing”; perhaps it could be replaced with “farming”

Line 33: replace “zone” with “zones”

Line 36: specify the two compartments

Introduction

Lines 57-59: sentence not clear, please rephrase it

Line 64: please replace with: “occupies the third place”

Line 65: omnivorous with a mostly herbivorous diet?

Line 68: please replace “rapid” with “high”

Line 70: please eliminate “prefers”

Line 78: tracts of the stomach? Did you intend portions or zones or regions?

Line 81-83: sentence not clear, please rephrase it

Materials and Methods

Line 97: north OF Italy

Lines 95-106: specify fish suppression method

Line 130: replace “section” with “sections”

Line 131: replace “field” with “fields”

Results

Line 151-152: : please use all verbs either in the past or in the present

Lines 159-163: in Figure 1 add the clear indications of the esophagus outlet and of the pyloric sphincter

Line 164: of THE esophagus.

Line 166: delete “it” and replace with “and”

Line 170: mucous or mucosa folds?

Line 171-172: sentence not clear, please rephrase it

Line 180: replace “portion” with “layer”

Line 182: please use all verbs either in the past or in the present

Lines 184-185: avoid repeating words: eliminate one of the “vessels”

Line 186: replace “portion” with “layer”; please further describe the circular structures

Line 190: raised or organized?

Lines 191-192: sentence not clear, please rephrase

Lines 193-194: ….and BY a muscular layer of striated muscle

Figure 2: indicate in figure A the squamous epithelium and the striated muscle layer

Line 196-197: again: mucous or mucosa folds?

Line 197: …a layer OF squamous epithelium; delete “a”

Figure 3: indicate the prismatic cells in figure B; indicate the third layer in figure E; in figure D the scale bar seems to be cut, can you verify it?

Line 201: …the tubular glands OF THE stomach

Line 204: specify what the third layer is

Lines 220-260: please add clear descriptions considering the figures (add arrows, asterisks etc to help the redear to better understand what you are describing.

Lines 231-232: avoid repeating words (covered)

Lines 247-248: sentence not clear, please rephrase

Line 253: eliminate the comma

Lines 257-258: please use all verbs either in the past or in the present

Line 261. 3.3. Morphometrical analysis – please add a statistical analysis to this chapter and then discuss the data again

Line 272: delete “it”

Line 291. “took place by” is not appropriate, please replace

Line 298. Figure 6: it should be better to indicate in the images what you describe in the text, mentioning for instance the differences in colors. Moreover, the figure should be re-organized.

Line 309: “even under” non-appropriately used, please replace; replace “scanning microscope” with “SEM”

Line 317: avoid repeating words (portion)

Figure 7E: indicate in the image the connective tissue

Discussion

Line 350: replace “for” with “of”; insert “a” before “digestive function”

Lines 360-363: revise the sentence taking into account the use and position of the commas

Lines 366-367: sentence not clear, please rephrase it

Line 367: replace “make up” with “constitute”

Lines 374-378: sentence too long and not clear

Line 390: “going” not appropriate, please rephrase

Line 398: it seems that a word is missing: “…would ensure a better? as already observed...”

Line 411-413, “To counteract this acidic environment, mucins of different pH are 411 secreted along all the secreting portions of the stomach. In this study through histochemistry we have shown mucin secretion at different pH.” Please rephrase the sentence in order to better explain the kind of mucins production.

Lines 422-423: sentence not clear, please rephrase

Line 424: it is not clear: the muscular longitudinal layer remains constant throughout the lumen? what do you mean?

Line 428: replace “in favor of” with “through”; what do you mean with “against gradient”?

Line 432: please use the articles: add “the” before “stomach” and be

Author Response

The authors of the study showed the presence of 5 different regions in the stomach of Nile tilapia that allowed them to formulate functional hypotheses for the digestive process that could be used for studies of different dietary plans. In general, the manuscript is interesting and I recommend it for publication after some revisions, such as: 1) the statistical analysis of the morphometric data, in order to give strength to the results; 2) the description of the histological images should be contextualized, where possible, on the images using arrow, asterisk, arrowhead, etc. in order to make reading the results clearer.

The authors are grateful to the reviewer for the careful review of the manuscript.

In red the responses to comments and suggestions are provided below

Abstract

Line 29: “growth”: higher growth rates? Done, see lines 41-42.

Line 30: avoid repeating the word “growing”; perhaps it could be replaced with “farming” Done, see line 42.

Line 33: replace “zone” with “zones” Done, see line 46.

Line 36: specify the two compartments. Done, see lines 48-49.

Introduction

Lines 57-59: sentence not clear, please rephrase it. Done, see lines 79-84.

Line 64: please replace with: “occupies the third place”. Done, see line 91.

Line 65: omnivorous with a mostly herbivorous diet? Authors deleted it.

Line 68: please replace “rapid” with “high” Done, see line 95.

Line 70: please eliminate “prefers” Done, see line 99.

Line 78: tracts of the stomach? Did you intend portions or zones or regions? Authors intended regions, see line 109.

Line 81-83: sentence not clear, please rephrase it. Authors deleted the sentence.   

Materials and Methods

Line 97: north OF Italy. Done, see line 132.

Lines 95-106: specify fish suppression method. Authors added it, see line 135.

Line 130: replace “section” with “sections” Done, see line 176.

Line 131: replace “field” with “fields” Done, see line 177.

Results

Line 151-152: please use all verbs either in the past or in the present. Done, see lines 206-210.

Lines 159-163: in Figure 1 add the clear indications of the esophagus outlet and of the pyloric sphincter. Figure 1 was changed.

Line 164: of THE esophagus. Done, see line 224.

Line 166: delete “it” and replace with “and” The sentence was rephrased, see lines 226-228.

Line 170: mucous or mucosa folds? It is mucosa, see line 232.

Line 171-172: sentence not clear, please rephrase it. The sentence was rephrased, see lines 231-237.

Line 180: replace “portion” with “layer” Done, see line 245.

Line 182: please use all verbs either in the past or in the present. The sentence was rephrased, see lines 247-249.

Lines 184-185: avoid repeating words: eliminate one of the “vessels” Done, see line 251.

Line 186: replace “portion” with “layer”; Done, see line 252.

Please further describe the circular structures. Done, see line 253.

Line 190: raised or organized? Done, see line 259.

Lines 191-192: sentence not clear, please rephrase. The sentence was rephrased, see lines 259-261.

Lines 193-194: ….and BY a muscular layer of striated muscle. Done, see line 263.

Figure 2: indicate in figure A the squamous epithelium and the striated muscle layer. Authors changed figure 2 and legend related.

Line 196-197: again: mucous or mucosa folds? Done, see line 268.

Line 197: …a layer OF squamous epithelium; delete “a” Done, see line 268.

Figure 3: indicate the prismatic cells in figure B; indicate the third layer in figure E; in figure D the scale bar seems to be cut, can you verify it? Authors changed figure 3 and legend related.

Line 201: …the tubular glands OF THE stomach. Legend was rephrased.

Line 204: specify what the third layer is Done, see line 277.

Lines 220-260: please add clear descriptions considering the figures (add arrows, asterisks etc to help the redear to better understand what you are describing. The authors have reformulated the paragraph, see lines 294-342.

Lines 231-232: avoid repeating words (covered). Done, see lines 307-308.

Lines 247-248: sentence not clear, please rephrase. Sentence was rephrased, see lines 327-328.

Line 253: eliminate the comma. Done, see line 333.

Lines 257-258: please use all verbs either in the past or in the present. Done, see lines 338-340.

Line 261. 3.3. Morphometrical analysis – please add a statistical analysis to this chapter and then discuss the data again. Done, see paragraph lines 348-363.

Line 272: delete “it” The whole paragraph was changed.

Line 291. “took place by” is not appropriate, please replace The whole paragraph was changed.

Line 298. Figure 6: it should be better to indicate in the images what you describe in the text, mentioning for instance the differences in colors. Moreover, the figure should be re-organized. Authors changed figure 6 and legend related.

Line 309: “even under” non-appropriately used, please replace; replace “scanning microscope” with “SEM” “Even under the scanning microscope was deleted, see line 431.

Line 317: avoid repeating words (portion). Done, see lines 440-441.

Figure 7E: indicate in the image the connective tissue. Authors changed figure 7 and legend related.

Discussion

Line 350: replace “for” with “of”; insert “a” before “digestive function”. Sentence was rephrased, see lines 485-488.

Lines 360-363: revise the sentence taking into account the use and position of the commas. Sentence was rephrased, see lines 499-504.

Lines 366-367: sentence not clear, please rephrase it. Sentence was rephrased, see lines 508-511.

Line 367: replace “make up” with “constitute”. Sentence was rephrased, see lines 508-511.

Lines 374-378: sentence too long and not clear. Sentence was rephrased, see lines 518-522.

Line 390: “going” not appropriate, please rephrase. Sentence was rephrased, see lines 535-538.

Line 398: it seems that a word is missing: “…would ensure a better? as already observed...” Authors added the word “digestion”, see line 546.

Line 411-413, “To counteract this acidic environment, mucins of different pH are 411 secreted along all the secreting portions of the stomach. In this study through histochemistry we have shown mucin secretion at different pH.” Please rephrase the sentence in order to better explain the kind of mucins production. Authors added a better explanation, see lines 565-577.

Lines 422-423: sentence not clear, please rephrase. Sentence was rephrased, see lines 586-587.

Line 424: it is not clear: the muscular longitudinal layer remains constant throughout the lumen? what do you mean? Sentence was rephrased, see lines 587-589.

Line 428: replace “in favor of” with “through”; what do you mean with “against gradient”? Sentence was rephrased, see lines 592-594.

Line 432: please use the articles: add “the” before “stomach” and be. Done, see line 603.

Reviewer 2 Report

In this paper entitled "A morphological and ultrastructural study of the anterior digestive tract of adult Nile tilapia Oreochromis niloticus" it was studied the morphology of the esophagus and the stomach of Tilapia, a fish species used in aquaculture. The authors identified  five distinct zones of the stomach(ZONE 1: esophagus-gastric lumen passage; ZONE 2: descending glandular portion; ZONE 3: fundic portion; ZONE 4: ascending glandular portion; ZONE 5: gastric-pyloric transition portion;) by using light and scanning microscopy.  In addition, by using histochemical methods they observed  the presence of neutral and acidic mucins along all the secreting portions of  the esophagus and stomach.  The authors hypotized that the acidic secretions seem to be necessary for the digestion of algae and detrital bacteria, while the neutral ones are responsible for protecting the mucosa from the acidity of the gastric contents, for facilitating the transit of food and the regulation of gastric pH. By using morphometrical analysis, the authors studied the thicknesses of the various layers of which Tilapia’s stomach is composed by evaluating superficial epithelium, glandular epithelium, submucosal, circular  and longitudinal musculature. They concluded that their description brings deeper knowledge of the morpho structure of the anterior digestive tract of Nile tilapia, a necessary condition for normal health and growth.

The topic of this study is interesting, however, it can be accepted after minor revision.

I have itemized my major concerns in the following paragraphs.

Material and methods

Morphometry:

It is necessary to specify the statistical analysis carried out for morphometry. In particular, the authors must add the description of the statistical tests used for morphometrical data (i.e. ANOVA , specify the p value, software, etc...). 

Figure 6

Add calibration bar in Panel A of this figure

Figure 5

Add the information regarding the statistical analysis in the figure legend in order to clarify the data in these graphs (i.e. the data are mean ± standard deviation, p value and then the significant values). 

Author Response

 The authors are grateful to the reviewer for the careful review of the manuscript.

In red the responses to comments and suggestions are provided below. Also, English language was revised.

Material and methods

Morphometry:

It is necessary to specify the statistical analysis carried out for morphometry. In particular, the authors must add the description of the statistical tests used for morphometrical data (i.e. ANOVA , specify the p value, software, etc...). 

At lines 178-185 authors described the statistical analysis carried out.

Figure 6

Add calibration bar in Panel A of this figure.

Authors changed figure 6 and legend related.

Figure 5

Add the information regarding the statistical analysis in the figure legend in order to clarify the data in these graphs (i.e. the data are mean ± standard deviation, p value and then the significant values). 

Authors changed figure 5 and legend related.

Reviewer 3 Report

The work presented by Palladino et al. on “A morphological and ultrastructural study of the anterior digestive tract of adult Nile tilapia Oreochromis niloticus” is interesting and enhance the present understanding of digestive function in the fish. However, I could see several flaws that need to be addressed. Overall, the English language is poor and needs to be improved.

My comments for improvement of the MS are noted below:

  1. Line 16-17: Line is not constructed properly. Rephrase.
  2. Line 64-65: Provide reference to the statement.
  3. Line 95 and elsewhere: Do not repeat the full scientific name after first mention. Genus can be abbreviated.
  4. Line 96: gram can be written as g or gm. Not “gr”.
  5. Line 101-102: were the tissues washed to remove blood, and adipose fats?
  6. Line 199-120: sentence not clear.
  7. Line 123: at what magnification?
  8. Line 136: CO2 should be corrected
  9. Line 164-168: the description in text if it could be explained in the Figure will be better.
  10. Figure 2, 3: more description and clarity required as in figure 4
  11. Figure 5C: Spelling mistake, similarly, D, E …Expand the heading
  12. Figure 6 shows neutral mucins. Indicate in footnote and mark in picture using arrows
  13. Discussion need to be precise and able to extend comprehensive knowledge for further work. Needs rewriting.

Author Response

The authors are grateful to the reviewer for the careful review of the manuscript.

In red the responses to comments and suggestions are provided below. Also, English language was revised.

My comments for improvement of the MS are noted below:

  1. Line 16-17: Line is not constructed properly. Rephrase.

We have reformulated the sentence, see lines 66-67.

  1. Line 64-65: Provide reference to the statement.

References 28-29 were added.

  1. Line 95 and elsewhere: Do not repeat the full scientific name after first mention. Genus can be abbreviated.

Done.

  1. Line 96: gram can be written as g or gm. Not “gr”.

Done.

  1. Line 101-102: were the tissues washed to remove blood, and adipose fats?

To remove blood we washed in PBS (line 139).

Around the organs we did not find enough visceral fat to require mechanical removal

  1. Line 199-120: sentence not clear.

Authors rephrased the sentences, see lines 145-148.

  1. Line 123: at what magnification?

Authors added it, see line 169.

  1. Line 136: CO2 should be corrected.

 Done.

  1. Line 164-168: the description in text if it could be explained in the Figure will be better.

Authors changed figure 2 and legend related and improved it.

  1. Figure 2, 3: more description and clarity required as in figure 4……

Authors changed figure 3 and legend related and the legend of figure 4.

  1. Figure 5C: Spelling mistake, similarly, D, E …Expand the heading.

Authors changed figure 5 and legend related.

  1. Figure 6 shows neutral mucins. Indicate in footnote and mark in picture using arrows.

Authors changed figure 6 and legend related.

  1. Discussion need to be precise and able to extend comprehensive knowledge for further work. Needs rewriting.

Authors rewrote both discussion and conclusions.

Round 2

Reviewer 3 Report

Well done. Authors have carefully responded the suggestions. The manuscript looks fine. No further comments.